# Antigen-Specific T Cell Immunotherapy Targeting Claudin18.2 in Gastric Cancer

**DOI:** 10.3390/cancers14112758

**Published:** 2022-06-02

**Authors:** Bo Xu, Fangjun Chen, Xin Zhang, Zhongda Wang, Keying Che, Nandie Wu, Lixia Yu, Xiangshan Fan, Baorui Liu, Jia Wei

**Affiliations:** 1The Comprehensive Cancer Centre of Drum Tower Hospital, Medical School of Nanjing University & Clinical Cancer Institute of Nanjing University, Nanjing 210008, China; yhtmyf@126.com (B.X.); fangjunchen07@163.com (F.C.); mg1935034@smail.nju.edu.cn (X.Z.); keyingche@163.com (K.C.); nandiewu@sina.com (N.W.); njyulixia@hotmail.com (L.Y.); 2Department of Gastroenterology, Nanjing First Hospital, Nanjing Medical University, Nanjing 210006, China; 3The Comprehensive Cancer Centre of Nanjing Drum Tower Hospital, Clinical College of Nanjing Medical University, Nanjing 210008, China; wzd199519@126.com; 4Department of Pathology, Nanjing Drum Tower Hospital, The Affiliated Hospital of Nanjing University Medical School, Nanjing 210008, China; fxs23@163.com; 5Collaborative Innovation Center for Personalized Cancer Medicine, Nanjing Medical University, Nanjing 210008, China

**Keywords:** gastric cancer, Claudin18.2, immunotherapy, peptide

## Abstract

**Simple Summary:**

Claudin18.2 is expressed in the primary and metastatic gastric cancer, making Claudin18.2 a suitable target for antigen-specific T cell immunotherapy. In this study, we first identified 12 Claudin18.2 peptides that had immunogenicity, and found that T cells stimulated by Claudin18.2 peptides had stronger anti-tumor activity and higher effective cytokine-secreting ability in vitro. We also found that Claudin18.2 peptide reactivity was associated with older age and higher Claudin18.2 expression, which helped to screen appropriate patients. The value of Claudin18.2 in the T cell-based GC immunotherapy has been affirmed in this study.

**Abstract:**

T cell-based immunotherapy has led to many breakthroughs in the treatment of solid tumors. In this study, we found that membrane protein Claudin18.2 was a promising antigen in T cell-based immunotherapy for gastric cancer (GC). Firstly, we identified five HLA-A*0201- and seven HLA-A*1101-restricted T cell epitopes of Claudin18.2. Peripheral blood mononuclear cells (PBMCs) stimulated by Claudin18.2 peptides showed progressive anti-tumor ability and higher effective cytokine secretion than unstimulated PBMCs in vitro. In total, 81.8% of GC patients were Claudin18.2-positive by immunohistochemical (IHC) detection, and a positive correlation between Claudin18.2 expression and peptide reactivity (*p* = 0.002) was found. Clinicopathological features analyses demonstrated that Claudin18.2 expression did not correlate with gender, age, stage or Lauren classification. Survival analysis showed that a longer median progression-free survival (mPFS) was not related to peptide reactivity (*p* = 0.997), but related to a lower Claudin18.2 expression level (*p* = 0.047). These findings establish a foundation for the clinical application of Claudin18.2 targeted T cell-based immunotherapy in GC.

## 1. Introduction

GC is the fifth most common cancer worldwide and contributes to a large number of cancer-related deaths [1,2]. The 5-year survival rate of advanced GC is less than 10% [3]. Most GC patients are diagnosed at an advanced stage, which makes GC difficult to treat [4]. For patients with incurable, recurrent, or metastatic gastric adenocarcinoma, the current first-line treatment is chemotherapy using mainly platinum and fluoropyrimidine [5]. However, with a poor 5–7-month median progression-free survival (PFS) and 9–11-month median overall survival (OS), the efficacy of chemotherapy is dismal [6]. Antibody therapies targeting VEGFR [7,8] and HER-2 [9,10,11] have shown modest benefits in GC clinical studies. Immune checkpoint inhibitors (ICIs), such as anti-PD-1 antibodies, are only useful in a minority of GC patients due to the complex tumor microenvironment (TME) [12,13,14,15]. Notable efforts have been focused on T cell-based immunotherapy in GC because of its promising application future. 

Claudin18.2 is a tight junction protein from the Claudin family. It is strictly expressed in differentiated gastric mucosa epithelial cells under physiological conditions and is pathologically expressed in different types of primary and metastatic cancers [16,17]. Claudin18.2 has been reported to be packaged within the tight junction supramolecular complex physiologically, which made it impossible to target therapeutically. However, the malignant transformation in cancer cells leads to the exposure of Claudin18.2 epitopes [18], making it an ideal target for targeted therapy. The retained expression of Claudin18.2 has been found in 77% of primary GCs and 66% of its lymph node metastases [16], which makes Claudin18.2 one of the most notable targets in GC studies. Zolbetuximab is a specific Claudin18.2 antibody that has shown effective anti-tumor ability in gastric, oesophageal and pancreatic cancer studies [19,20]. A phase II clinical trial (FAST; NCT01630083) has shown that patients with ≥2+ membrane staining intensity of Claudin18.2 in ≥40% tumor cells could benefit from zolbetuximab therapy [21]. In addition, Claudin18.2-specific chimeric antigen receptor engineered T (CAR-T) cells have shown partial or complete tumor elimination in GC patient-derived tumor xenograft (PDX) models [22]. However, in a Claudin18.2-specific CAR-T phase I clinical trial for advanced gastric and pancreatic adenocarcinoma patients, the total objective response rate was only 33.3%, and cytokine release syndromes were observed during the treatment [23]. Thus, T cell-based therapies that are safer and more efficient are needed.

Here, we demonstrate a relatively simple and safe Claudin18.2 peptide-based T cell therapeutic modality for GC. T cells showed obvious immune responses after stimulation by Claudin18.2 peptides. We verified the high Claudin18.2 expression rate in GC through IHC detection and found a positive correlation between Claudin18.2 expression and peptide reactivity. Finally, a longer mPFS was found to be related to a lower Claudin18.2 expression level. These findings established an experimental foundation for the clinical application of Claudin18.2 peptide-specific T cells in GC.

## 2. Materials and Methods

### 2.1. Patients, Specimens and Ethical Statement

A total of 44 advanced GC patients were selected and 29 of them were identified with the targeted HLA types (HLA-A*0201 and HLA-A*1101) and their autologous PBMCs were used to analyze T cell responses to Claudin18.2 peptides. To test Claudin18.2 expression, formalin-fixed, paraffin-embedded (FFPE) tissue specimens from all the 44 patients were collected. None of the patients had radiotherapy, chemotherapy or other medical intervention before specimen collection. The study was approved by the Ethics Committee of Nanjing Drum Tower Hospital. Written informed consent was obtained from each subject.

### 2.2. HLA Typing

PCR-sequence-based typing (CapitalBio Technology, Beijing, China) of peripheral blood samples were performed to identify common Asian HLA types (HLA-A*0201 and HLA-A*1101). Briefly, the target HLA genes were amplified by PCR, the amplified products were purified and tested by Western blot, and then Sanger sequencing was performed. The sequencing results were analyzed to determine HLA type. 

### 2.3. Epitope Prediction and Peptide Synthesis

The NetMHCpan 4.0, SYFPEITHI, IEDB and NetCTLpan 1.1 tools were used to predict MHC class I binding of 9-mer and 10-mer Claudin18.2 peptides to HLA-A*0201 and HLA-A*1101 alleles. Peptides with an IC50 less than 500 nM or a %rank less than 2.0 were identified as MHC binders. Peptide synthesis and purification were performed at ChinaPeptides.

### 2.4. T Cell Response Analysis 

Ficoll density gradient centrifugation was used to isolate PBMCs from heparinized blood samples. For this, 1 × 10^5^ PBMCs were incubated with a corresponding peptide (25 μM) in 200 μL culture medium in each U-bottomed well. The culture medium for PBMCs consisted of AIM-V (Gibco, Life Technology Invitrogen, Waltham, MA, USA), 10% fetal calf serum (FCS, Gibco, Waltham, MA, USA), and Interleukin-2 (IL-2, 100 U/mL, PeproTech, Cranbury, NJ, USA). Half of the culture medium was changed with fresh culture medium containing peptide (25 μM) and IL-2 (100 U/mL) every 3 days. After 3 cycles of peptide stimulation (9 days), the entire amount of culture medium in each well was replaced by AIM-V medium containing peptide (25 μM) only. After 24 h, on day 10, the specific T cells responded to each peptide were evaluated by IFN-γ cytometric bead array (BD Biosciences, San Jose, CA, USA) and ELISPOT array (Dakewei, Beijing, China). No-peptide (media only) was used as a negative control and phytohemagglutinin (PHA) stimulation was used as a positive control.

### 2.5. Cytokine Cytometric Bead Array Analysis

Cytokine concentrations in the culture supernatants were measured by cytometric bead array (BD Biosciences, USA). Human IFN-γ Flex Set (Bead B8, BD Biosciences, USA) was used to detect IFN-γ. Human Th1/Th2 Cytokine Kit II (BD Biosciences, USA) was used to detect 6 cytokine protein levels in a single sample: Interleukin-2 (IL-2), Interleukin-4 (IL-4), Interleukin-6 (IL-6), Interleukin-10 (IL-10), Tumor Necrosis Factor-α (TNF-α), and Interferon-γ (IFN-γ). The samples were tested using an Accuri C6 flow cytometer (BD Biosciences, USA) and analyzed using FCAP version 3.0 array software (Soft Flow, USA).

### 2.6. Cytotoxicity Assay

Carboxy fluorescein succinimide ester/propidium iodide (CFSE (Invitrogen, USA)/PI (Sigma-Aldrich, Germany)) labeling cytotoxicity assay was used to test the tumor-cell-killing ability of antigen-stimulated T cells. GC cell lines with HLA-A*0201 were used as target cells. Target cells were labeled with 4 mM CFSE for 10 min at 37 °C in phosphate buffer solution (PBS). A 10-fold volume of PBS was used to wash the cells 3 times and stop the labeling. CFSE-labeled tumor cells were then incubated with T cells (with or without peptide stimulation) at different effector/target ratios for 6 h. PI was added to each well for 15 min at 4 °C to determine the cell death ratio. Samples were analyzed by flow cytometry.

### 2.7. Generation of Dendritic Cells (DCs) and Peptide Reactive T Cells

DCs were generated by plate adherence of PBMCs. Briefly, 5 × 10^6^ cells/mL PBMCs were incubated in AIM-V medium for 2 h at 37 °C, 5% CO_2_. Non-adherent cells were collected and used to culture T cells. The adherent cells were collected and cultured with CellGro DC media (CellGenix, Germany) containing 1% human serum (collected and processed in-house), GM-CSF (800 IU/mL), and IL-4 (1000 IU/mL) to prepare DCs. After 96 h, DCs were harvested and pulsed with peptides (25 μM) individually for approximately 4 h at 37 °C. Then, peptides pulsed with DCs were incubated with T cells at a ratio of 1:5 in AIM-V medium supplemented with 10% FCS (Gibco, USA), IL-2 (100 U/mL), IL-7 (10 ng/mL), and IL-15 (10 ng/mL). Fresh complete medium was added every 2–3 days. On day 7, the proportion of peptide specific T cells was assessed by flow cytometry or ELISPOT assay. According to the growth of peptide specific T cells, irradiated K562 were cocultured with T cells for re-stimulation. Artificial antigen presenting cells (APCs) K562 were transduced with HLA-A*1101 or HLA-A*0201 molecular, and loaded with HLA-A*1101 or HLA-A*0201 restricted Claudin18.2 peptides. (The K562 cells expressing CD137L, CD80, and HLA-A*1101/HLA-A*0201 were constructed by our library.)

### 2.8. Claudin18.2 Expression

Claudin18.2 expression was determined by IHC staining in FFPE tissues from GC patients. Briefly, tissue slides were incubated with anti-Claudin18.2 antibody (Abcam, Cambridge, UK) at a 1/800 dilution overnight at 4 °C and then rewarmed for at least 5 min. Slides were washed three times with PBS. After the slides were dry, secondary antibody was added drop-wise to cover the tissue entirely, and the slides were incubated in a 37 °C incubator for 40 min. Nuclei were stained with hematoxylin. Samples showing any specific staining with ≥1+ intensity were defined as Claudin18.2 positive. ≥40% of tumor tissues had specific staining with ≥2+ intensity was defined as high expression (according to FAST trial). The staining intensity was classified into 4 grades: no membrane or cytoplasmic reactivity as 0; weak membrane or cytoplasmic reactivity as 1+; moderate membrane or cytoplasmic reactivity as 2+; and strong membrane or cytoplasmic reactivity as 3+. The percentage of Claudin18.2-positive cells was calculated as the estimated number of Claudin18.2-positive cells divided by the estimated overall number of tumor cells in each sample [16,17,24].

### 2.9. Statistics SPSS Was Used for All Statistical Analyses

GraphPad Prism 7.0 was used to graph the data. The results were expressed as mean ± standard error of the mean (SEM). Data samples were compared using *t* tests, and Pearson’s chi-square test. A *p* value of less than 0.05 was considered significant. Survival analyses were performed using the Kaplan–Meier method and compared with a log-rank test.

## 3. Results

### 3.1. Claudin18.2 Peptides Specifically Induced Immune Activation in GC Patients

In total, 12 9-mer to 10-mer Claudin18.2 peptides were predicted to bind to HLA-A*0201 and HLA-A*1101 molecules using 4 programs with different algorithms [25]; the amino acid sequences for those 12 peptides are listed in Table 1. The personalized immunogenicity identification of Claudin18.2 peptides were performed in vitro on 29 advanced GC patients with matched HLA types (21 were HLA-A*0201, 12 were HLA-A*1101, 4 of them were HLA-A*0201 and HLA-A*1101 heterozygotes). The secretion of the effector cytokine IFN-γ were measured using both Cytometric bead array and ELISPOT assay after PBMCs were stimulated in vitro with individual peptides for 10 days. A three-fold or greater increase in IFN-γ secretion compared to the negative control group was considered significant. As a result (Table 1), 52.4% (11/21) of HLA-A*0201 subtype patients had reactivity to the five predicted HLA-A*0201 peptides and among them, peptide 118–126 showed the highest response rate (63.6%, 7/11) and was selected for subsequent experiments. In total, 50% (6/12) of HLA-A*1101 subtype patients had reactivity to the seven predicted HLA-A*1101 peptides. The response rate for each peptide is shown in Table 1. In addition, nine of the responded patients had multiple peptides reactivity, indicating Claudin18.2 peptides had a general immunogenicity in GC patients.

### 3.2. Claudin18.2 Peptide Stimulated T Cells Showed Promising Anti-Tumor Ability In Vitro 

PBMCs from HLA-A*0201 patients were firstly used to test the reactivity to peptide 118–126 by 10 days co-stimulation in vitro; if greater than three-fold IFN-γ secretion was detected, the PBMCs were then used to generate 118–126 peptide-specific T cells. Here, we chose patient 005, whose PBMCs showed a 3.24-fold increased IFN-γ secretion after 10 days 118–126 peptide stimulating (Figure 1A,B) to generate 118–126 peptide-specific T cells. To verify peptide reactivity of the generated T cells, IFN-γ secretion was tested after two cycles of 118–126 peptide stimulating. Here, 3.15-fold (cytometric bead array, Figure 1C,D) and 4.71-fold (ELISPOT assay, Figure 1E,F) IFN-γ secretion was detected, respectively, which substantiated the reactivity of generated T cells to Claudin18.2 118–126 peptide. Next, three types of Claudin18.2-positive HLA-A*0201 GC cell lines (NUGC4, AGS, and KE39, Appendix A) were used to test the anti-tumor ability of 118–126 peptide-specific T cells by CFSE/PI labeling cytotoxicity assay. In all three GC cells, 118–126 peptide-specific T cells showed prior anti-tumor ability (*p* < 0.05) (Figure 2A–C), which indicated that Claudin18.2 epitopes were processed and presented on tumor cells and could be recognized by our Claudin18.2 peptide-specific T cells. The culture supernatants of the highest effector/target ratio groups were collected to test IL-2, IL-4, IL-6, IL-10, TNF-α, and IFN-γ protein levels. The cytokine concentrations, especially TNF-α and IFN-γ, were significantly higher in the 118–126 peptide-stimulated T cell groups compared with non-stimulated groups (*p* < 0.001) (Figure 2D–I), demonstrating that Claudin18.2 peptide-stimulated T cells had stronger anti-tumor ability in vitro.

### 3.3. Claudin18.2 Highly Expressed in GC

A total of 44 FFPE samples of GC underwent IHC detection to measure Claudin18.2 expression (Appendix A). Of the 44 samples, 36 (81.8%) were Claudin18.2-positive, and among them 17 (47.2%) were 1+, 13 (36.1%) were 2+, and 6 (16.7%) were 3+. The distribution of overall Claudin18.2 positive cells’ percentage was as follows: 27 (61.4%) were 0–25%, 9 (20.5%) were 26–50%, 5 (11.4%) were 51–75% and 3 (6.8%) were 76–100%. Thus, the IHC results showed a relatively high Claudin18.2 expression rate in GC. For the 44 patients that had histological testing for Claudin18.2 expression, 34 (77.3%) were male, 21 (47.7%) were ≥60 years old, at the time of initial diagnosis, 6 (13.6%) were classified as stage I-II, 25 (56.8%) were stage III, 11 (25%) were stage IV, 2 (4.5%) were not capable to obtain pathological information. We explored the characteristics of Claudin18.2 expression in these 44 patients and found that Claudin18.2 expression situation was not correlated with gender, age, TNM stage, or Lauren classification (Appendix A). Median PFS data was available in 25 (56.8%) cases and a longer mPFS was found to be related to a lower Claudin18.2 expression level (*p* = 0.047, Figure 3A,B). 

### 3.4. Claudin18.2 Peptide Reactivity Was Higher in Claudin18.2 High Expressing GC Patients

Of the 29 patients sampled, 23 (79.3%) of the patients were male, 17 (58.6%) were ≥60 years old, at the time of initial diagnosis, 2 (6.8%) were classified as stage I-II, 13 (44.8%) were stage III and 14 (48.3%) were stage IV (Table 2). Claudin18.2 peptides had a higher reactivity in elderly (≥60) GC patients (*p* = 0.01, Table 2). Furthermore, Claudin18.2 peptide reactivity was positively correlated with Claudin18.2 expression level (*p* = 0.002, Table 2). However, Claudin18.2 peptide reactivity showed no apparent relationship with gender, TNM stage, or Lauren classification (Table 2). Median PFS data was available in 20 (69%) cases and no significant relationship was found between Claudin18.2 peptide reactivity and mPFS (Figure 3C).

## 4. Discussion

Antigen-specific cytotoxic T lymphocytes (CTLs) that were induced with tumor epitope-based peptides had been widely used in the treatment of malignant solid tumors. Antigen-specific CTLs had the potential to induce and boost effective anti-tumor responses. So far, T cells targeting EBV [26], KRAS [27], MYD88 [28], HER-2 [29] and other antigens had been identified by peptide stimulation. In addition, tumor-infiltrating lymphocytes (TILs) could also be used for inducing and sorting antigen-specific T cells. TILs were thought to be easier to obtain antigen-specific T cells since they had already been stimulated by antigens within the tumor tissue [30]. However, due to the difficulty in obtaining TILs, PBMCs that were easy to obtain were selected in this study. Instead of adding extra peptide-pulsed DCs, APCs such as DCs and activated macrophages in the originally added PBMCs and the irradiated K562 cells were used as peptide-pulsed stimulator cells; all of them had strong survival and antigen presenting ability in vitro [25].

Recently, Claudin18.2 had become a star target in GC due to its favorable expression characteristics. Clinical trials had been carried out on Claudin18.2-targeted drugs and CAR-T cells. In phase II clinical studies for Zolbetuximab (IMAB362) [19,21], a positive correlation between Claudin18.2 expression and the therapeutic benefit was observed. This was consistent with our findings that Claudin18.2 peptides could induce higher T cell reactivity in GC patients with high Claudin18.2 expression. This suggested that Claudin18.2 expression could be used to predict immunotherapy efficacy and screen for patients that were highly likely to benefit. Claudin18.2 was a cell surface molecule, which facilitated its use as a target for CAR-T therapy. Several phase I clinical trials for Claudin18.2 targeted CAR-T cells was carried out [22,23,31]. The total objective response rates were only 33.3–48.6%, which revealed that Claudin18.2-targeted CAR-T therapy was very promising in patients with Claudin18.2-positive cancers of the digestive system. Besides, those clinical trials further confirmed the value of Claudin18.2 in T cell-based immunotherapy, not only CAR-T therapy, but also antigen-specific CTLs and even future engineered T cell receptor T cell (TCR-T) therapy. Varying degrees of adverse reactions were observed during the treatment in Claudin18.2 targeted CAR-T clinical trials. Not like CAR-Ts which need gene transfection, our peptide-stimulated antigen-specific T cells were safer and cost less time and money. On the other hand, our findings were also valuable in the preparation of Claudin18.2 vaccines. Overall, our founding demonstrated a low-cost and low-risk treatment model which needed no gene transfection, and on the other hand, with promising efficacy and safety potential, over 50% of patients had reactivity to Claudin18.2 peptides, and peptide-stimulated T cells showed remarkable anti-tumor ability. 

Neoantigen was regarded as the cornerstone of immunotherapy; however, shared common neoantigen in GC was relatively lacking compared with other malignant cancers such as lung cancer and intestinal cancer. Our team had constructed a shared neoantigen peptide library of solid tumors earlier [25], which provided a pattern for a timely and efficient identification of neoantigen in clinical T cell-based immunotherapy. Based on the pattern, a clinical trial (NCT03171220) of antigen-specific peptide-based personalized immunotherapy in solid tumors were conducted, and had encouraging clinical outcomes. We identified that Claudin18.2 was highly expressed in GC and had promising immunogenicity, making it appropriate for individualized antigen-based GC immunotherapy. By combining with the constructed peptide library, the application of Claudin18.2 targeted therapy would be more extensive. Antigen-specific T cells tend to have high PD-1 expression, the combination of antigen-specific T cells and anti-PD-1 therapy is of great value.

Since the patient samples collected in this study were all from the past 2 years, only PFS data could be achieved. The results showed that a longer median PFS was related to a lower Claudin18.2 expression level while not associated with peptide reactivity. This did not mean that the peptide reactivity of T cells did not contribute to disease control; in fact, PFS varied in each patient and the longest PFS were from peptide-reactive group (Appendix A). Considering the differences in disease stages, genetic backgrounds, treatment strategies and TME, only the spontaneous T cell response to Claudin18.2 seemed not sufficient enough to contribute to survival. On the other hand, there were some immune escape mechanisms existing in vivo to limit the anti-tumor ability of Claudin18.2 antigen-specific T cells; for example, the up-regulated expression of PD-L1 on tumor cells and PD-1 on antigen-specific T cells in peripheral blood. We found 9 of the collected patients had multiple Claudin18.2 peptides reactivity; however, due to the possible reasons listed above, no significant difference of mPFS between the multiple-peptide reactive group and the single-peptide reactive group (*p* = 0.591) had been found (Appendix A). Claudin18.2 expression and its immunogenicity alone were not sufficient enough to contribute to survival, especially for these patients who did not have Claudin18.2-targeted therapy or specific T cell therapy. Claudin18.2 peptides could specifically induce immune activation, which was of great significance for the guidance of treatment. One other notable result in our study was the observation that peptide reactivity was higher in elderly GC patients. This might be because tumors in those elderly patients were less malignant and they had relatively strong immune systems. The observation that antigen reactive T cells-inducing capabilities of those peptides were diverse and varied among patients might be attributed to the genetic differences of the selected patients; this suggested that it was necessary to verify the peptide reactivity individually.

The tremendous potential of epitope-based peptide vaccines in treating malignant solid tumors was discovered in [32,33]. A successful peptide vaccine must be comprised of immunogenic epitopes to increase immune responses in immune cells, particularly CTLs [34], which are the most effective component of the immune system capable of destroying tumor cells. The value of Claudin18.2-targeted T cell-based immunotherapy in GC is confirmed in this study, and the immunogenicity of Claudin18.2 affirmed in our study made it also a suitable target for vaccine therapy. A total of 12 different MHC-restricted Claudin18.2 peptides were determined to have various degrees of immunogenicity and were conducive to the subsequent development of peptide vaccines against GC.

## 5. Conclusions

In summary, the value of Claudin18.2 in GC immunotherapy had been affirmed in this study. The immunogenicity of Claudin18.2 peptides, which were found for the first time, provided bases for the clinical application of Claudin18.2 targeted T cells. The correlation between the reactivity of Claudin18.2 peptides and Claudin18.2 expression would help to screen appropriate patients in future antigen-specific T cell-based clinical trials.

## Figures and Tables

**Figure 1 cancers-14-02758-f001:**
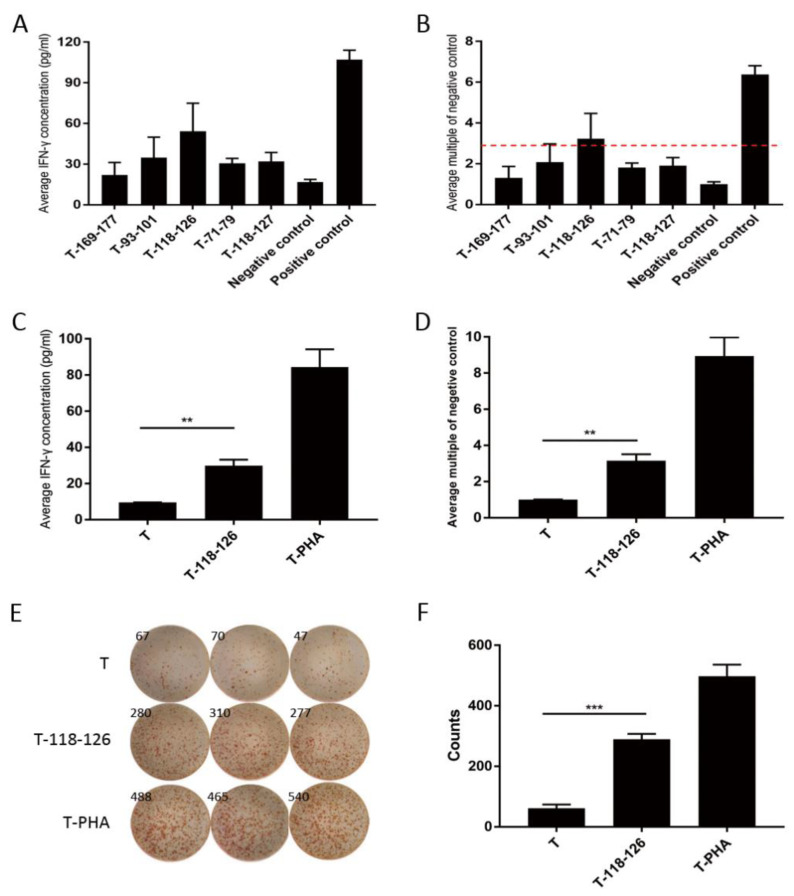
Peptide 118–126-stimulated T cells showed stronger IFN-γ secretion ability. (**A**) The average concentration of IFN-γ (pg/mL) after 10 days’ stimulation with each HLA-A*0201 restricted Claudin18.2 peptides for patient 005. (**B**) The average multiples of increased IFN-γ secretion compared to negative control after 10 days stimulation with each HLA-A*0201 restricted Claudin18.2 peptides for patient 005, who showed the best reactivity to peptide 118–126. (**C**) The average concentration of IFN-γ (pg/mL) secreted by generated 118–126 peptide-specific T cells after 2 cycles of stimulation (cytometric bead array). (**D**) The average multiples of increased IFN-γ secretion compared to negative control after 2 cycles of 118–126 peptide stimulation (cytometric bead array). (**E**) ELISPOT assay results of PBMCs stimulated by 118–126 peptide after 2 cycles of stimulation. (**F**) Statistical chart of positive spot counts from ELISPOT assay results. T—no peptide negative control; T-118–126—peptide 118–126-stimulated T cells group; T-PHA—positive control; **—*p* < 0.01; ***—*p* < 0.001.

**Figure 2 cancers-14-02758-f002:**
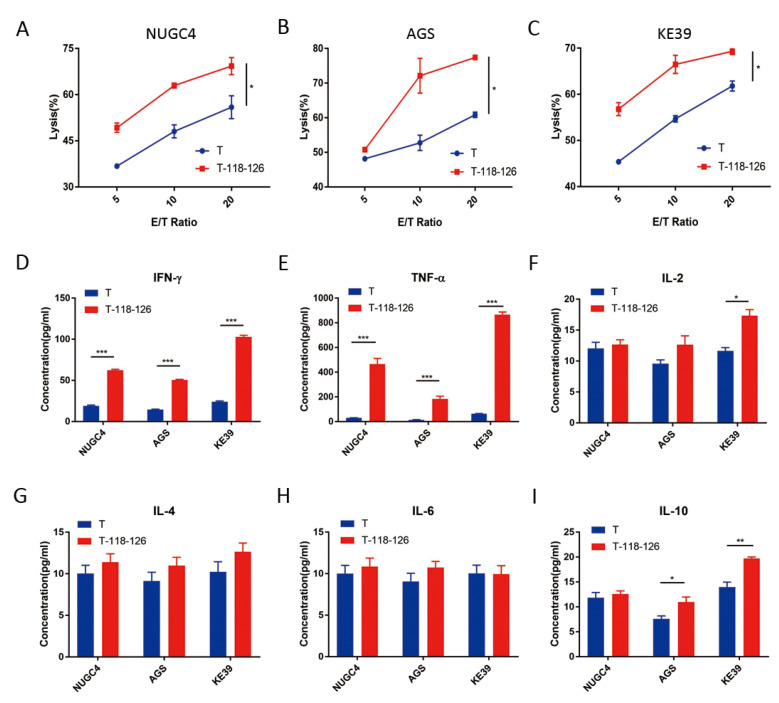
Claudin18.2 peptide-stimulated T cells showed prior anti-tumor ability in vitro. (**A**–**C**) Peptide 118–126-stimulated T cells showed prior anti-tumor ability in 3 GC cell lines (NUGC4, AGS and KE39) by CFSE/PI labeling cytotoxicity assay. (**D**–**I**) The concentration of IFN-γ, TNF-α, IL-2, IL-4, IL-6 and IL-10 in the culture supernatants of the 20:1 ratio groups. T—no peptide negative control; T-118–126—peptide 118–126-stimulated T cells group; *—*p* < 0.05; **—*p* < 0.01; ***—*p* < 0.001.

**Figure 3 cancers-14-02758-f003:**
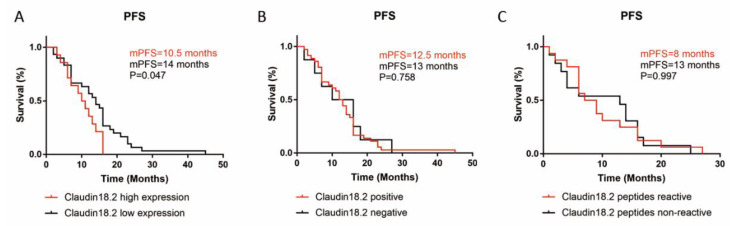
The relation of mPFS with Claudin18.2 expression or peptide reactivity in GC patients. (**A**) mPFS was related to Claudin18.2 expression level. (**B**) Claudin18.2 positive or negative expression had no relation with mPFS. (**C**) mPFS had no relation with Claudin18.2 peptide reactivity. High expression—≥40% of tumor tissues had specific Claudin18.2 staining with ≥2+ intensity; Low expression—no Claudin18.2 expression or <40% of tumor tissues had specific Claudin18.2 staining; *p* < 0.05 was considered statistically significant.

**Table 1 cancers-14-02758-t001:** Construction of HLA-A*0201 and HLA-A*1101 restricted Claudin18.2 peptides.

HLA Type	Peptide Name	Peptide Sequence	HLA-Binding Affinity (%Rank)				Reactive Patients	Rate
			NetMHCpan 4.0	SYFPEITHI	IEDB	NetCTLpan 1.1		
HLA-A*0201	169–177	YTFGAALFV	0.169	20	0.8	1	4	0.364
	93–101	GLLVSIFAL	0.186	28	0.5	0.3	4	0.364
	118–126	TLTSGIMFI	0.127	23	1	0.8	7	0.636
	71–79	GLPAMLQAV	0.173	26	0.28	1	3	0.273
	118–127	TLTSGIMFIV	0.672	19	0.51	1	3	0.273
HLA-A*1101	228–236	STGFGSNTK	0.091	23	0.42	0.8	3	0.500
	94–102	LLVSIFALK	0.446	20	0.86	0.3	1	0.167
	218–226	VAYKPGGFK	0.131	10	0.56	0.8	2	0.333
	217–226	SVAYKPGGFK	0.106	24	0.33	0.4	4	0.667
	42–51	AVFNYQGLWR	0.371	26	0.82	0.8	3	0.500
	212–221	HASGHSVAYK	0.483	12	0.73	1	2	0.333
	93–102	GLLVSIFALK	1.123	24	0.94	0.4	3	0.500

‘Reactive patients’ refers to the number of patients who were tested to be reactive to each peptide. ‘Rate’ indicates reactive patients for each peptide divided by peptide-reactive patients for each HLA type.

**Table 2 cancers-14-02758-t002:** Characteristics of Claudin18.2 peptide reactivity in GC patients.

Factors		Total Number	Reactive	Not Reactive	*p*-Value
Gender	Male	23	11	12	0.525
	Female	6	2	4	
Age	≥60	17	11	6	0.01
	<60	12	2	10	
TNM stage	II	2	1	1	0.824
	III	13	5	8	
	IV	14	7	7	
Lauren classification	Intestinal type	8	5	3	0.429
	Diffuse type	5	2	3	
	Missing	16	8	8	-
Claudin18.2 expression	High	9	8	1	0.002
	Low and negative	11	2	9	
	Missing	9	3	6	-

‘Missing’ includes cases where classification was not applicable or not assessable. *p* < 0.05 was considered statistically significant.

## Data Availability

The data that support the findings of this study are available on request from the corresponding author.

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
