# Peer review of "Antigen-Specific T Cell Immunotherapy Targeting Claudin18.2 in Gastric Cancer"

_cancers, 2022, doi:10.3390/cancers14112758_

Round 1

Reviewer 1 Report

Immune checkpoint inhibitors (ICIs) including pembrolizumab, nivolumab, durvalumab, atezolizumab, etc. have been recently evaluated in gastric cancer patients. Despite ICI seem to have finally found their role in gastric cancer, several questions remain unanswered. Among these, the lack of validated biomarkers of response represents an important issue since only a proportion of patients benefit from immunotherapy. Based on these premises, a greater understanding of the role of potential biomarkers including programmed death ligand 1 (PD-L1) expression, tumor mutational burden (TMB), microsatellite instability (MSI) status, gut microbiota and several others is fundamental. In addition, clinical trials on gastric cancer immunotherapy widely differed in terms of drugs, patients, designs, terms of study phases, and inconsistent clinical outcomes.
Based on these premises, the study assesses a current, timely topic.
We recommend some changes:
- We believe this article is suitable for publication in the journal although major revisions are needed. The main strengths of this paper are that it addresses an interesting and very timely question and provides a clear answer, with some limitations. 
- The background of the changing scenario of medical treatment in gastric cancer should be better discussed, and some recent papers regarding this topic should be included ( PMID: 33916915 ; PMID: 33916206 )

- A linguistic revision is needed.
Major changes are necessary.

Reviewer 2 Report

This is overall a very interesting study regarding new and promising treatment for gastric cancer. I believe that the material collected for the study should regard new patients, since only PFS data can be achieved from older samples. However, the conclusions of the study could vouch for a position among other relative studies alrerady published in Cancers. I suggest minor revisions of the manuscript.

Round 2

Reviewer 1 Report

Acceptance.
